# Characterization of Dissolved Organic Matter from Agricultural and Livestock Effluents: Implications for Water Quality Monitoring

**DOI:** 10.3390/ijerph20065121

**Published:** 2023-03-14

**Authors:** Guizhi Qi, Borui Zhang, Biao Tian, Rui Yang, Andy Baker, Pan Wu, Shouyang He

**Affiliations:** 1Key Laboratory of Karst Georesources and Environment, Guizhou University, Ministry of Education, Guiyang 500025, China; 2School of Biological, Earth and Environmental Sciences, UNSW Sydney, Sydney, NSW 2052, Australia; 3College of Resources and Environmental Engineering, Guizhou University, Guiyang 500025, China; 4Guizhou Karst Environmental Ecosystems Observation and Research Station, Ministry of Education, Guiyang 500025, China

**Keywords:** DOM, fluorescent components, EEMs-PARAFAC, surrogate for water quality, livestock effluents, agricultural effluents

## Abstract

There is growing concern about the impact of agricultural practices on water quality. The loss of nutrients such as nitrogen and phosphorous through agricultural runoff poses a potential risk of water quality degradation. However, it is unclear how dissolved organic matter (DOM) composition is associated with pollution levels in water bodies. To address this, we conducted a cross-year investigation to reveal the nature of DOM and its relationship to water quality in agricultural effluents (AEs) and livestock effluents (LEs). We discovered that DOM fluorescence components of AEs were mainly from autochthonous and terrestrial sources, while in LEs it was primarily from autochthonous sources. LEs showed a higher β:α and biological index (BIX) than AEs, indicating that LEs had higher biological activity. Compared to the LEs, DOM in AEs exhibited a higher humification index (HIX), illustrating that DOM was more humic and aromatic. Overall, our results suggest that the BIX and fluorescence index (FI) were best suited for the characterization of water bodies impacted by LEs and AEs. Excitation–emission matrix spectroscopy and parallel factor (EEMs-PARAFAC) analysis showed that DOM in AEs was mainly a humic-like material (~64%) and in LEs was mainly protein-like (~68%). Tryptophan-like compounds (C1) were made more abundant in AEs because of the breakdown of aquatic vegetation. The microbial activity enhanced protein-like substances (C1 and C2) in LEs. Our study revealed a positive correlation between five-day biochemical oxygen demand (BOD_5_) concentrations and tyrosine-like substance components, suggesting that fluorescence peak B may be a good predictor of water quality affected by anthropogenic activities. For both LEs and AEs, our results suggest that peak D may be a reliable water quality surrogate for total phosphorus (TP).

## 1. Introduction

Agricultural pollution caused by many chemicals (e.g., fertilizers, plastic films, pesticides, herbicides, farm feeds, etc.) has greatly contributed to the decline in water quality and aquatic ecosystems. The China Rural Statistical Yearbook revealed that the effective usage rates of fertilizer and pesticide are less than 1/3, and the mulch recycling rate is lower than 2/3. In addition, the effective treatment rate of livestock and poultry manure is less than 50%, leading to the gradual eutrophication of water bodies, which is a serious matter of concern (Ministry of Agriculture,2019). Agricultural wastewater has become a major water and soil pollution source in China [1]. It not only causes the degradation of agroecosystems [2] but also destroys biodiversity and ecosystem balance. The major impacts of livestock farming on freshwater systems are as follows: nutrient inputs (especially nitrogen and phosphorus) [3], high solid contents of organic wastewater, pesticides [4], and bacterial and protozoan pathogens [5]. Intensively managed livestock [6] and stockpiling of food and feed in farming areas are likely to be sources of contaminants that can end up in freshwater systems [7].

DOM in Earth’s aquatic ecosystems is a heterogeneous carbon-hydrogen mixture of amino, aliphatic, and aromatic functional groups containing oxygen, nitrogen, and sulfur compounds [8,9]. It is considered the main reactive carbon pool [10]. DOM concentrations and molecular compositions vary both globally and locally, this being attributed to various sources of pollution in aquatic ecosystems [11]. It is also vulnerable to human activities and climate change [12]. Therefore, it plays a crucial role in the biogeochemical cycling of aquatic ecosystems, affects metal complexation and transport [13,14], nutrient supply [12,14], and phytoplankton and microbial activity [15]. In addition, DOM is an indispensable indicator of water quality in Earth’s aquatic ecosystems [16,17].

EEMs are a fast and responsive optical method and a practical tool for predicting water quality [16,18]. The use of fluorescent EEMs combined with parallel factor analysis (PARAFAC) is a valuable approach for detecting the origin of DOM in varied aquatic environments [19,20]. DOM in surface water can be influenced by individual biological activities such as agriculture, domestic sewage, livestock farming, etc. Thus, it is closely related to local water pollution [21]. These human activities strongly influence the DOM composition and biogeochemical behavior in aquatic ecosystems. The alteration in DOM composition poses new challenges for monitoring and characterizing water quality in differently impacted water bodies using EEM. Moreover, few relationships are known between EEMs-PARAFAC fluorescence components and water quality in AEs and LEs. Therefore, this study aimed to develop the use of three-dimensional fluorescence to track DOM generated from AEs discharged from agricultural fields and farm discharges from us. The specific objectives here were to (1) characterize the fluorescence of AEs and LEs under different levels of pollution, (2) investigate the seasonal variation in DOM, and (3) reveal DOM in response to the water quality of the AEs and LEs.

## 2. Material and Methods

### 2.1. Description of the Experimental Setup

The research area was situated in the educational laboratory base of Guizhou University (Figure 1), which is situated about 3 km north of the city center of Huaxi District, Guiyang City, Guizhou Province (106°43′ E, 26°36′ N). The experimental area occupies about 660 × 10^3^ square meters, including 430 × 10^3^ square meters of arable land, 130 × 10^3^ square meters of the green forest belt, 50 × 10^3^ square meters of the orchard, 40 × 10^3^ square meters of a fish pond, and 50 × 10^3^ square meters of the nursery. The sampling site has a huge arable land, planted with many fruit-bearing trees and surrounded by a residential area. The temperature in the study area varied between 0 to 30 °C, with the highest temperature in September and the lowest in January. The mean monthly rainfall was between 2 to 343 mm, experiencing maximum and minimum rainfall in June and December, respectively. Rainfall mobilizes nutrients from terrestrial sources into water bodies, affecting the water quality as well as influencing the biogeochemical and hydrological processes [22]. Upstream of the AEs, there is also a great deal of cultivated and free-range fowl, such as chickens and ducks, which results in anthropogenic sources of organic matter. Domestic and livestock sewage are the main sources of LEs.

### 2.2. Sample Collection and Analytical Procedure

To characterize the dissolved organic matter of AEs and LEs, two sampling sites were established at the educational laboratory base of Guizhou University (Figure 1). One of the sampling sites was a pond to collect the AEs. The other sampling point was a ditch collecting LEs. The AEs and LEs were sampled for one year, from October 2020 to September 2021, and the collection frequency was kept twice a month, with a total of 24 sets of water samples collected. Dissolved oxygen (DO) was assessed on-site by a portable water quality parameter (Multi 3630 IDS SET G, GRE).

Routine water quality analysis of filtered samples was performed in the laboratory. The total nitrogen (TN) concentration was determined using the alkaline potassium persulfate ablation UV spectrophotometric technique (HJ 636-2012), and the ammonium molybdate spectrophotometric method (GB 11893-89) was used to test the TP content. In addition, the salicylic acid spectrophotometric method (HJ 536-2009) was applied to measure the ammonia nitrogen (NH_3_-N) content. The HACH BOD Trak II (HACH, Loveland, CO, USA) pressure difference method was used to determine the BOD_5_ of water samples. The potassium dichromate method (GB 1191489) was used to test the COD content. A TOC V-CPH elemental analyzer (Jena Analytica, Thuringia, Germany) was used to examine the DOC concentrations.

Pre-acid-washed plastic containers were used to take water samples from AEs and LEs at a depth of around 10–20 cm. On the sampling day, the water samples were filtered using 0.45 μm syringe filters. They were put into acid-washed HDPE bottles for storage after filtering. After which, the bottles were moved with ice and kept at 4 °C in the dark pending additional testing. Within 48 h of collecting the water sample, the EEM fluorescence spectra were measured. LEs have a high concentration of DOC, so the three-dimensional fluorescence spectrum was measured to ensure the accuracy of the experimental results after diluting the water sample 1000 times. A three-dimensional fluorescence spectrum scan of the filtered water samples was performed using a Horiba Agualog fluorescence spectrometer (Agualog-UV-800-C, Piscataway, New Jersey, USA). The experimental temperature was adjusted to 20 °C with a Thermo Scientific ARCTIC SC100-A10 Refrigerated Circulator. Other parameters were as follows: the scanning speed: 12,000 nm·min^−1^, scanning range: excitation wavelength (Ex): 240–800 nm with a scanning interval of 3 nm, emission wavelength (Em): 250–800 nm with a sweeping interval of 1 nm; both excitation and emission slit widths were set to 1 nm. The scanning spectra were automatically calibrated using the instrument. Millipore ultrapure water (resistivity 18.2 MΩ·cm) was used as a blank, and Rayleigh scattering and Raman scattering coefficients were automatically deducted using the analyzer.

### 2.3. Statistical Analysis

To detect the fluorescence components in the DOM, the DOMFluor toolbox in MATLAB 2018b [23,24] was utilized. The drEEM toolbox’s smoothing function was utilized to reduce residual scattering induced via EEM [25]. Fitting the PARAFAC model with 2 to 8 components yielded the proper number of components. Split-half analysis, residual analysis, and random initialization analysis [24] were used to derive the final fluorescence component model [23]. The four-component fluorescence model used in this study was confirmed and accounted for 97.8% of the dataset’s variance. Validation was carried out using the model available in the OpenFluor website’s online spectrum library (http://www.openfluor.org, accessed on 6 September 2022). A Tucker consistency better than 0.95 was established when compared to previously discovered equivalent PARAPAC components [25]. For the DOM, the following optical indices were determined: (i) FI, BIX, and HIX that reflect the DOM source [16,26]; and (ii) a (355) and Fn (355) signify chromophoric DOM (CDOM) and fluorescent DOM (FDOM) concentrations [16], respectively. The nonparametric Mann–Whitney U test was used to evaluate water quality factors with seasonal fluctuations in EEM fluorescence [27]. These analyses were performed using SPSS 18.0 and were significant at *p* < 0.05. Mantel text analysis was performed using R4.04 (‘ggcor’package) and correlation plots were drawn.

## 3. Results and Discussion 

### 3.1. Water Chemistry

The seasonal variations in the water quality indices including DO, TP, TN, NH_3_-N, COD, and BOD_5_ are depicted in Figure 2 There is no discernible seasonal variation between the DO of AEs, which varies from 3.78 to 14.78 mg/L with large oscillations in the northern hemisphere during autumn, and the LEs, which range from 0.01 to 0.09 mg/L with large changes in the summer.

TP consists of organic and inorganic phosphorus and is less than 0.1 mg/L in surface water [28]. Higher concentrations of TP come from residential and commercial water discharges as well as drainage from fertile agricultural land. In AEs, TP ranges from 0.1 to 0.4 mg/L. In summer, over 50% of the samples exceeded the quality standard of 0.3 mg/L, while in autumn and winter, all values were below this level. With considerable seasonal fluctuation (*p* < 0.01) between autumn and other seasons, LEs’ TP varied from 14.99 to 259.90 mg/L, exceeding the V standard of the People’s Republic of China’s national standards for surface water quality standards (GB3838-2002-V) by a wide margin (0.4 mg/L). Higher values were recorded at the discharge point in spring and summer, due to a mixed source of leaching, the runoff of urine and manure from the farm, and the influence of domestic drainage. For TN, AEs varied from 0.27 to 5.24 mg/L throughout the year, with a mean value of 1.39 mg/L and no significant seasonal differences. Two observations were found to be higher than GB3838-2002-V (<2 mg/L) in spring, which was related to the variation in monthly rainfall distribution and the discharge of untreated live sewage during the study period. The annual variation in LEs ranged from 211.81 to 8732.29 mg/L, with a mean value of 3727.51 mg/L, which far exceeded GB3838-2002-V, with seasonal differences. The observed values of TN were significantly higher in summer than in other seasons, which was related to the changes in monthly rainfall distribution and production activities of the farm during the study period. Low concentrations of the water-soluble gas NH_3_N (0.1 mg/L) can be found in untreated natural water, and the NH_3_N of AEs and LEs come from nitrogenous organic matter and the gas exchange between water and the atmosphere, microbial degradation, inputs from households and agricultural inputs, and the influence of anthropogenic activities. The NH_3_N of LEs ranged from 262.34 to 8389.47 mg/L, well above GB3838-2002-V (<2 mg/L).

COD and BOD_5_ are two parameters used to quantify the organic pollution load. The AEs’ COD values ranged from 12 to 364 mg/L, with significant seasonal differences between summer and other seasons (*p* < 0.01), and more than 70% of the samples exceeded GB3838-2002-V (<40 mg/L) in summer. BOD_5_ for AEs ranged from 1.8 to 10 mg/L, with significant seasonal differences between autumn and other seasons (*p* < 0.01). The COD values of LEs ranged from 3650 to 17,350 mg/L, and all samples’ values exceeded GB3838-2002-V (<10 mg/L), with significant seasonal differences between winter and other seasons (*p* < 0.01), and higher values were monitored in winter. The BOD_5_ values of LEs ranged from 1030 to 7800 mg/L, with significant seasonal differences between spring and other seasons (*p* < 0.01). The higher COD values compared to BOD_5_ indicate that a portion of organic material is degradable. 

When comparing AEs and LEs, we found that LEs had significantly higher levels of COD, BOD_5_, NH_3_-N, TP, and TN while having lower levels of DO. In our study, the level of TP, TN, BOD_5_, and NH_3_-N in AEs did not exceed GB3838-2002-V, while COD exceeded the standard, indicating that AEs were mainly polluted by organic matter. All livestock effluent’s water quality parameters exceeded GB3838-2002-V. The color of the water body was gray-black, indicating that the LEs were heavily polluted. PCA analysis was undertaken to estimate the statistical significance of the water quality parameters of AEs and LEs. In Figure 3A, it can be indicated that the first two principal components (PC1: 76%, and PC2: 11.5%) comprised ~87.5% of the overall data variance. COD, BOD_5_, NH_3_-N, TP, and TN showed larger positive loadings in the first major component, while DO had higher negative loadings. Most of the LE samples were found to be PC1 positive, with fewer PC2 loadings. In contrast, the samples from AEs were positively correlated with PC1. The PCA findings illustrate significant variations in the water quality parameters between AEs and LEs. Thus, we presumed that a different water quality also influenced the fluorescence properties of DOM [29].

### 3.2. Seasonal Variations in DOC, CDOM, and FDOM Concentrations

In AEs and LEs, seasonal fluctuations in DOC, CDOM, and FDOM contents were recognized (Figure 3B). DOC levels in AEs ranged from 2.99 to 37.15 mg/L, with a mean of 21.81 mg/L. The mean DOC content in LEs spanned from 497.62 to 6029.75 mg/L, with a mean value of 2255.00 mg/L. The DOC concentrations in AEs and LEs did not differ significantly due to seasonal variations. The DOC of LEs was much higher in autumn than other seasons, while for AEs it was highest in spring. The CDOM of AEs ranged from 2.35 to 12.23 m^−1^, with a mean value of 5.21 m^−1^, and there was a substantial seasonal change in the summer (*p* < 0.05). The CDOM of LEs ranged from 5.22 to 30.36 m^−1^, with a mean value of 11.16 m^−1^, and there was no seasonal difference. CDOM values were higher in spring than in other seasons. FDOM concentrations were expressed as Fn (355) and AE concentrations ranged from 0.37 to 2.07 a.u., with a mean value of 0.81 a.u. and with seasonal differences (*p* < 0.05). CDOM and FDOM concentrations were higher in summer than in other seasons, which was related to the increased surface runoff input due to increased rainfall. The FDOM of LEs ranged from 0.28 to 8.39 a.u., with a mean value of 3.03 a.u. and no significant seasonal variation. The intense human disturbances in the peri-urban regions may be the cause of the limited association between the DOC concentrations of AEs and LEs throughout the year (Figure 3). Similar to this, there is little association between the DOC and FDOM year-round. This shows that a significant part of the DOC in AE and LE water bodies is not chromophoric or fluorescent, just a low molecular weight of sugars, neutrals, etc. [30].

### 3.3. EEM-PARAFAC of DOM

#### 3.3.1. Fluorescent PARAFAC Components of DOM

Two protein-like chemicals (C1 and C2) and two humic-like substances (C3 and C4) were produced from EEM-PARAFAC to create four distinct fluorescent components of DOM (Table 1). Table 1 also displays the relevant excitation and emission wavelengths of these fluorescent fractions. We compared our findings with data from the previously published PARAFAC model [25] using the OpenFluor online spectrum library, and found a good match (Tucker agreement > 0.95 for Ex and Em spectra; Table 1). C1 is likely to be a protein-like substance (tryptophan-like) with fluorescence peak B [31,32]. C2 resembles the fluorescence peak T of protein-like substances (tyrosine-like) [33,34]. Peaks B and T are associated with aromatic ring amino acid structures, mainly due to enzymes produced during bacterial decomposition or a large number of proteins in biological debris [26]. C3 is similar to the fluorescence peak D of soil-enriched material [35,36]. C4 is consistent with fluorescence peak C of terrestrial humic material [32,37].

#### 3.3.2. Fluorescence Intensity of DOM

To analyze the seasonal effects more clearly on fluorescence components, we used the fluorescence intensity (Fmax) of two water bodies to study the seasonal variation in DOM component concentrations instead (Figure 4). Due to the various fluorescence quantum yields of various substances, the Fmax cannot be used to directly calculate the concentration of DOM fractions, but it may be used to calculate the relative content of each fraction [41]. Specifically for the class, the percentages of each fluorescence proportion of AEs were 18.06% (C1), 17.69% (C2), 26.10% (C3), and 38.15% (C4). We found that the Fmax values of each fraction were lower in autumn than in other seasons, which was related to the input of nutrients carried by rainfall. The Fmax of C2 (protein-like substances) in AEs was greater in the spring than in other seasons, and the highest value also occurred in the spring, which was associated with the drainage of runoff from rain and the dilution effect created in the summer. C1, C3, and C4 differed from other seasons in summer (*p* < 0.05), and C2 did not differ seasonally. C1 (tryptophan-like) is associated with microbial activity and can be transferred into the system (allochthonous) or generated by microbial and biological activity inside the system (autochthonous). Aquatic plants such as algae in the AEs’ water body in this research flourished in the spring and summer and decomposed in the winter, and their decomposition led to enhanced C1 in the spring and summer. In AEs, D and C peaks were observed to be more intense in spring than in other seasons, which is related to the drainage of stormwater runoff and the dilution effect produced in summer. The fluorescence intensity of LEs was significantly higher than that of AEs due to excessive organic pollutant loading. The percentages of each fluorescent component of LEs were 38.44% (C1), 30.44% (C2), 20.30% (C3), and 11.02% (C4), and were mainly protein-like substances (~68%). Due to the high COD in the exogenous input LEs, microorganisms may break down humic-like DOM in both aerobic and anaerobic environments [16,42], resulting in the higher fluorescence intensity of protein-like substances (C1 and C2) in LE samples. We discovered increased amounts of C1 and C2 protein-like substances in the winter as well, since C1 and C2 in LEs are thought to be microbial byproducts and the fluctuation in their fluorescence intensity is connected to microbial metabolism. The proportion of tryptophan-like (C2) substances in LEs was approximately twice that of AE, indicating that LEs are influenced to a greater extent by anthropogenic activities. C3 and C4 are both terrestrial and the soil represented by C3 contains fulvic acid, which is a typical natural substance [43]. We found that C4 was observed in lower concentrations in winter in AEs and higher concentrations in winter in LEs. C4 is derived from terrestrial sources of humic-like substances and macrophytes contribute poorly to the abundance of these components in natural water, with some humic-like substances associated with microbial activity. From Section 3.3.3, it is clear that the microbial activity of LEs is stronger and, therefore, this could be the source of the higher C4 observed in winter.

#### 3.3.3. Fluorescence Indices

The origination and humification of AEs and LEs were tracked using fluorescence indices, such as FI, BIX, HIX, and β:α (Figure 5). FI is commonly used to determine the origin of DOM [44]. Allochthonous sources, which include terrestrial sources, are marked by lower FI values (<1.4), in contrast to autochthonous sources, which include algal sources and/or microbial DOM, which normally lead to higher FI values (>1.9) [45,46]. FI values for all AE samples in our study ranged from 1.04 to 1.52, indicating that both terrestrial and microbial sources of DOM were contributing to the total DOM pool [16]. In terms of seasonal variation, the contribution to the DOM pool was mainly by autochthonous sources in the fall and winter seasons. Ninety percent of the samples contained AEs, ranging from 0.95 to 1.37, indicating that DOM derived from terrestrial sources made up most of the total DOM contribution. In addition, several samples had high FI values (~1.9), possibly due to an increase in the protein-like material associated with direct human activities. 

The BIX indicates an autochthonous source of DOM when it is >0.8 [16,47]. All AE samples in this study had BIX values >0.8, while 95% of LE samples had BIX values >0.8, indicating that DOM in this study adds an autochthonous input to the microbial activity. In general, DOM in AEs is largely autochthonous and a terrestrial source of microbial activity. Yet, DOM in LEs is primarily allochthonous and has strong microbial contribution. In natural aquatic environments, HIX typically ranges from 1.5 to 9.0, with higher readings indicating more DOM decay [26,48]. The HIX values of all AEs samples were within this range and were less decayed than water bodies non-impacted by agricultural and livestock effluents. The values were related to weakly decayed DOM and native biological activity [44].

β:α reflects the proportion of newly produced components in the overall DOM. β:α values are in proportion to the relative contribution of microbially produced DOM [16]. The HIX of less polluted AEs was higher. At the same time, the BIX was lower than that of the LEs, indicating that microbial activity enhanced the biological activity of LEs. Further, decaying crop residues in the water column increased the humification of AEs. The β:α of more polluted AEs was high, suggesting that it mainly consisted of newly formed components that were less humified than the AEs. However, there was no significant statistical distinction in the β:α values between AEs and LEs due to the input of yellow humus-like material and humus as well as nutrient induction. The increased FI values in the LEs compared to the less contaminated AEs indicated an increase in microbial production. This is because LEs promotes microbial activity and enhances microbial by-products, such as protein- and humic-like substances. The values of the BIX/FI, HIX/FI, and HIX/BIX of AEs showed regular variations along with the seasons (e.g., Figure 5). The BIX/FI decreased, while the HIX/FI increased with the season. The BIX/HIX also decreased with the season. These findings highlighted that LEs are influenced by the intensity of microbial activity, which is adapted in spring and summer. Therefore, the FT/BIX and HIX/BIX decreased with the season. In autumn and winter, microbial activity is restricted, leading to the decreased biological activity of the water column and increased decay [20,49]. The fluorescent components in AEs were complex and derived from various sources. Generally, these results shows that the BIX and FI could be used to characterize water bodies impacted by LEs and AEs.

### 3.4. Implications for Water Quality Monitoring

It is well known that conventional water quality parameters, with COD, BOD_5_, NH_3_-N, TP, and TN, were applied in water quality monitoring requirements that required much more time and cost. These parameters are also difficult to measure immediately in the environment at a high frequency in situ study. Fluorescence spectroscopy has become a potent laboratory or field method for the detection and characterization of DOM in water, with applications in effluent quality monitoring, pollution incident investigation, and routine on-site testing [50]. Can fluorescent components be substituted as a surrogate for the conventional water quality parameters test in water quality monitoring? CDOM is a crucial parameter that is tightly related to the nutrients quantified in the environment [51]. As Figure 6 shows, the partial mantel test showed that inorganic indicators (IC) of water quality parameters in the AEs were correlated with CDOM (0.2 < r < 0.4, 0.01 < *p* < 0.05). A similar correlation was not found in the LEs. No correlation was found between organic indicators (OC) and fluorescence peaks in both AEs and LEs. However, we found a correlation between protein-like substances (PL) and BOD_5_ in both AEs and LEs (0.2 < r < 0.4, *p* < 0.01; r > 0.4, *p* < 0.01), which indicates that protein-like substances are associated with microbial activity. AE and LE mantel tests showed that water quality parameters with different pollution levels were correlated with the fluorescence peaks. The intrinsic link between the fluorescence peaks and water quality parameters was difficult to address, which posed a challenge for monitoring water quality using fluorescence. The correlation between fluorescence and conventional water quality parameters needs to be further revealed at the different pollution levels of the effluents. Therefore, we calculated the correlation coefficients between the fluorescence peaks, selected DOMs (FDOM, CDOM), and water quality parameters (DOC, COD, TP, TN, NH_3_-N, and BOD_5_). Pearson’s correlation coefficients (r) showed that TP was significantly correlated with protein-like fluorescence peaks B (r = 0.82, *p* < 0.01) and T (r = 0.77, *p* < 0.001), and humus-like fluorescence peaks C (r = 0.83, *p* < 0.001) and D (r = 0.85, *p* < 0.001) in drainage from organically contaminated farmland. BOD_5_ was significantly correlated with protein-like fluorescence peak B (r = 0.59, *p* < 0.05). In heavily contaminated LEs, BOD_5_ was correlated with protein-like fluorescence peak B (r = 0.59, *p* < 0.05), the T peak (r = 0.48, *p* < 0.01), and D peak (r = 0.63, *p* < 0.05). TP and the D peak (r = 0.53, *p* < 0.05) were correlated. In both AEs and LEs, we found a positive correlation between BOD_5_ and the B peak and TP and the D peak, suggesting that the B peak may be a promising substitute for BOD_5_ and that the D peak may be a prospective substitute for TP.

This investigation did not observe a significant correlation between the PARAFAC component fluorescence peaks and parameters such as TN and NH_3_-N. The possible reason could be that a single parameter is insufficient to accurately portray the relationship between PARAFAC fraction, TN, and NH_3_-N. On the other hand, dissolved inorganic and particulate nitrogen might contribute significantly to TN and NH_3_-N. Due to different pollution levels, AEs and LEs exhibited different COD concentrations and specific regional tryptophan-like components. The Fmax of both COD and tryptophan-like components was higher in the LEs than in the AEs. There was also a significant difference between the Fmax of the PARAFAC fraction of AEs and LEs (U-test, *p* < 0.05). The fluorescence intensity of AEs was dominated by high levels of humic-like components D and C (Table 1), which corresponds with the preceding study [18]. It was likely related to the input of TP and TN agricultural pollutants. The fluorescence intensity of LEs was dominated by high levels of B and T, likely to be linked with primary productivity. In the present study, the correlation coefficients showed a significant positive correlation between CDOM, TP, and NH_3_-N in the AEs contaminated with organic pollutants. In contrast, no correlation was found in the absence of LEs, suggesting that TP and NH_3_-N contribute differently to the fluorescence fraction of CDOM, which is influenced by anthropogenic activities (fertilizer use, turning over the soil, and pesticide use).

## 4. Limitations

The relationship between water quality and fluorescence peaks may be inconsistent in time and space. In this study, we obtained data from October 2020 to September 2021, and the results of the study only found a relationship between some of the fluorescence peaks and water quality parameters. The chlorophyll content of the water samples was not measured in this study due to the experimental conditions, and thus the contamination analysis using WQI was not used. However, many factors influence the correlation between water quality fluorescence peaks. Topographic conditions such as land use, landscape composition and configuration, spatial scale, weather conditions, and slope may also affect the correlation between water quality fluorescence peaks. The above factors are all key issues in the future work of studying the correlation between water quality fluorescence peaks.

## 5. Conclusions

In this project, a year of monitoring was performed to better understand the correlation between DOM to water quality in AEs and LEs. The results show that four DOM compositions were detected via the EEMs-PARAFAC. AEs and LEs included two proteins-like compounds (C1: tryptophan-like substances and C2: tyrosine-like substances) and two humic-like substances (C3: fulvic acid-like substances and C4: terrestrial humic substances). The DOM in AEs was mainly humic-like material (~64%), which derived from endogenous and terrestrial inputs, while protein-like (~68%) was the dominant DOM, sourced from endogenous contribution in LEs. From a seasonal view, aquatic plants’ decomposition led to the enhancement of tryptophan-like substances (peak B) in AEs. The microbial activities increased protein-like substances (B and T) in LEs, which was associated with primary productivity. Comparing LEs with AEs, the former indicated higher biological activity, and the latter was more humic and aromatic. A significant correlation was observed between DOM composition, fluorescence indices, and conventional water quality parameters in two effluents with different pollution levels. The BIX and FI could be used for the characterization of water bodies impacted by LEs and AEs. Notably, we found a positive correlation between fluorescence peak B to BOD_5_ concentration and tyrosine-like substance components, suggesting that fluorescence peak B may be a good predictor of water quality affected by anthropogenic activities. A positive correlation between the TP concentration and humic-like (fulvic acid-like) substance components implies that fluorescence peak D was possibly a potential surrogate for organic phosphorus in water quality monitoring.

## Figures and Tables

**Figure 1 ijerph-20-05121-f001:**
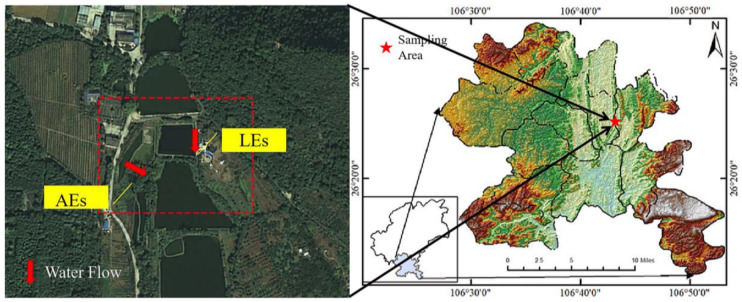
Location map of the study area and water sampling sites in the educational laboratory base of Guizhou University.

**Figure 2 ijerph-20-05121-f002:**
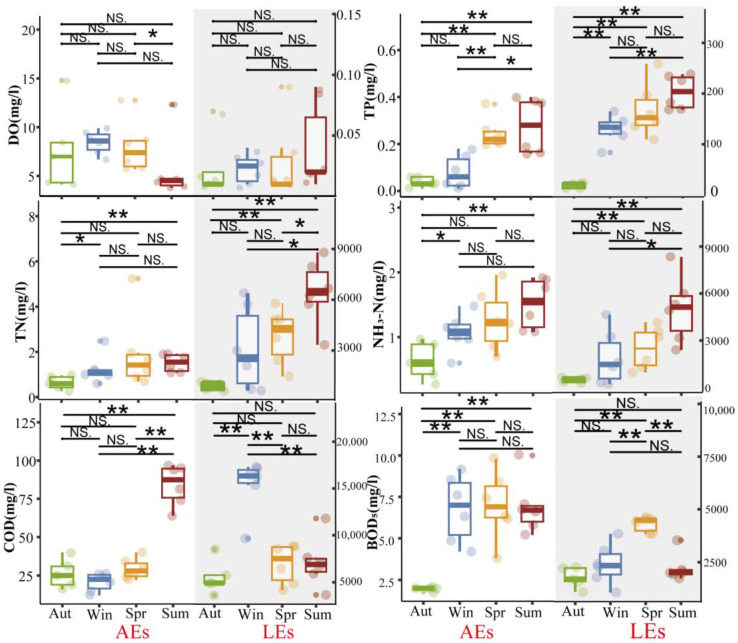
Seasonal variation in water quality parameters in AEs and LEs, seasonal difference U-test for AEs and LEs; * represents *p* < 0.05, ** represents *p* < 0.01. NS. stands for no seasonal variation.

**Figure 3 ijerph-20-05121-f003:**
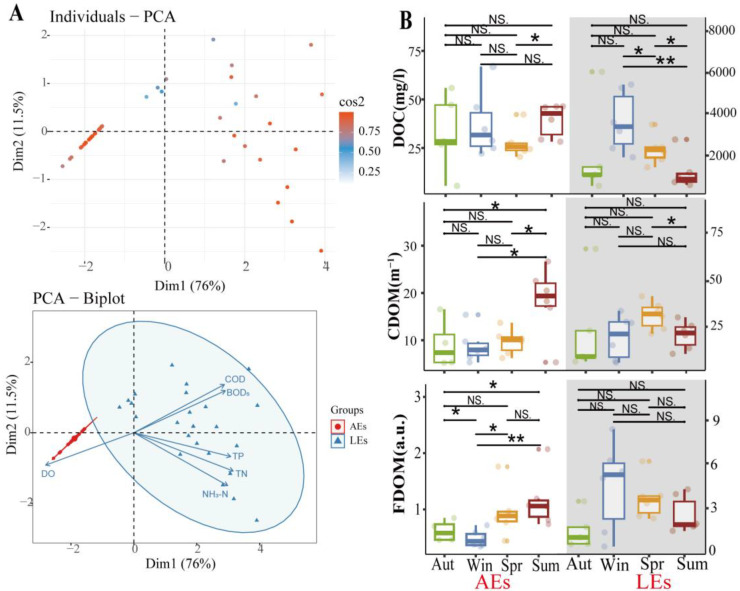
(**A**) Principal component analysis (PCA) of water quality parameters (TN, TP, COD, DO, BOD_5_, and NH_3_-N) of AEs and LEs; (**B**) Seasonal variation in DOM parameters in AEs and LEs, seasonal difference U-test for AEs and LEs; * represents *p* < 0.05, ** represents *p* < 0.01. NS. stands for no seasonal variation.

**Figure 4 ijerph-20-05121-f004:**
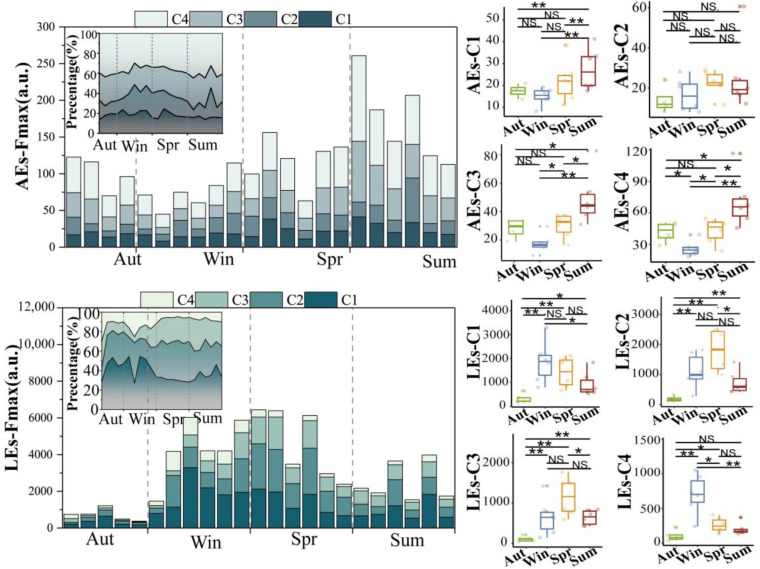
Seasonal variation in fluorescence component Fmax and seasonal difference U-test for AEs and LEs; * represents *p* < 0.05, ** represents *p* < 0.01. NS. stands for no seasonal variation (the figure shows the Fmax of LEs as a result of a 1000-fold dilution.).

**Figure 5 ijerph-20-05121-f005:**
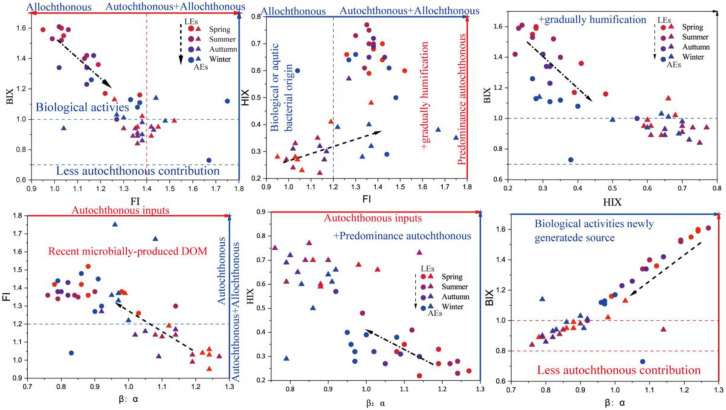
Relationship between FI, BIX, HIX, and β:α in AEs and LEs and variations with the season.

**Figure 6 ijerph-20-05121-f006:**
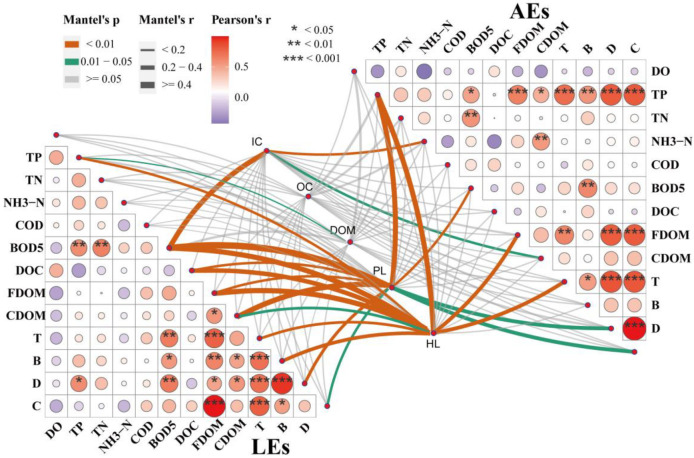
The relationships between conventional water quality parameters (inorganic indicators (IC), organic indicators (OC)), DOM, and fluorescence peaks (protein-like peaks (PL) and humus-like peaks (HL)) were observed in AEs and LEs. The partial Mantel test was applied using the ‘ggcor’ R package. Mantel’s r (coefficient of correlation) indicates the correlation, in which the width of the line denotes the level of correlation and the color of the lines represents the *p*-value. A color and asterisk gradient show Spearman’s correlation coefficients and the statistical significance, respectively.

**Table 1 ijerph-20-05121-t001:** EEM positions, EEMs contours, spectral loadings, and loadings of fluorescent components of AEs and LEs determined using the EEM-PARAFAC model and the corresponding fluorescence peaks.

Model	Refence Model λEx/λEm (Ref.)	Probable Fluorescence Peak	Components, EEM Contours, Spectral Loadings, and Location		Description
C1: 272/354	C7: 279/348 C3: 273/352 C8: 270/360	T	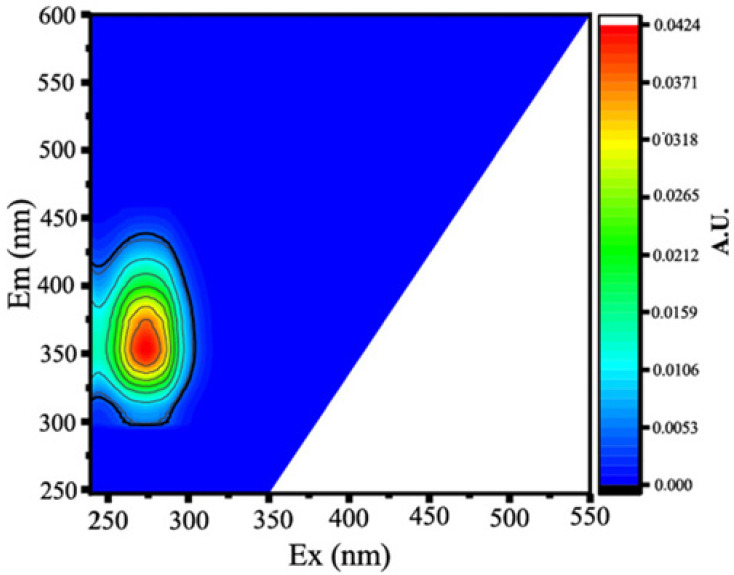	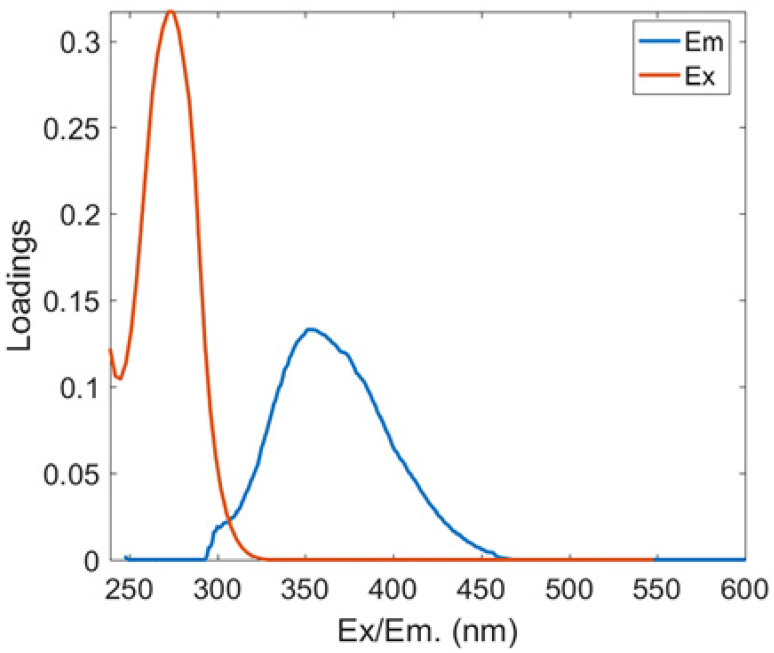	Protein-like free tryptophan derived from autochthonous processes
C2: 272/302	C3: 266/302 C5: 277/326 C5: 274/318	B	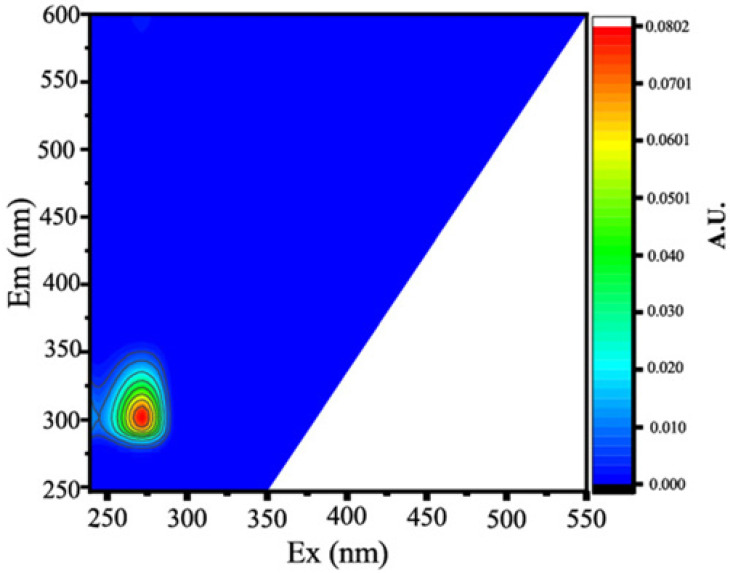	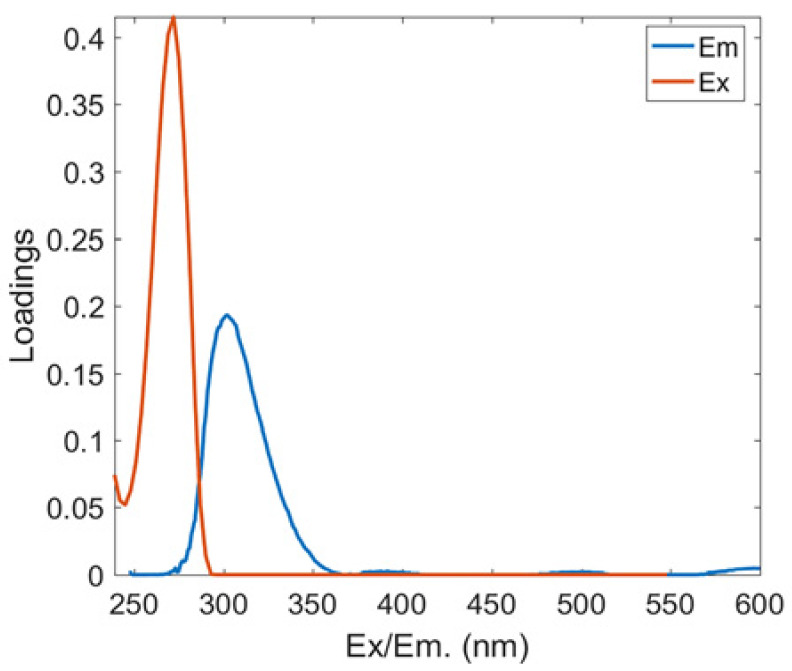	Protein-like free tyrosine-like moieties attributed to microbial origin
C3: 308/393	C2: 241(307)/398 C2: 302/400	D	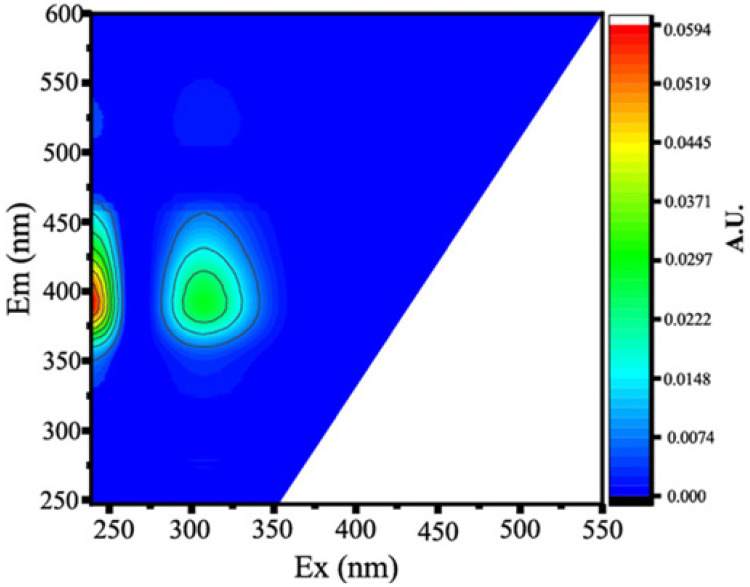	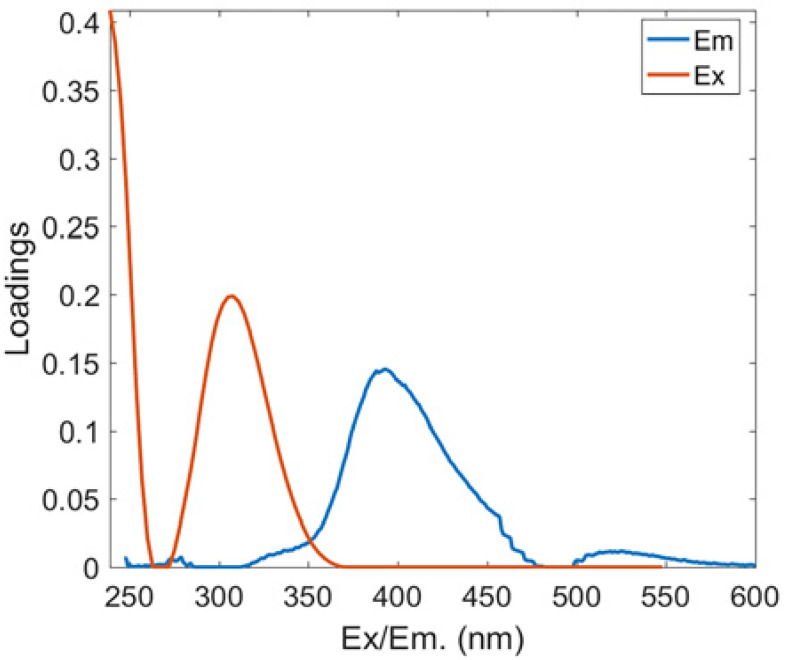	Humic-like, Fulvic-like, similar to semiquinone molecules
C4: 263/442	C2: 258/470	C	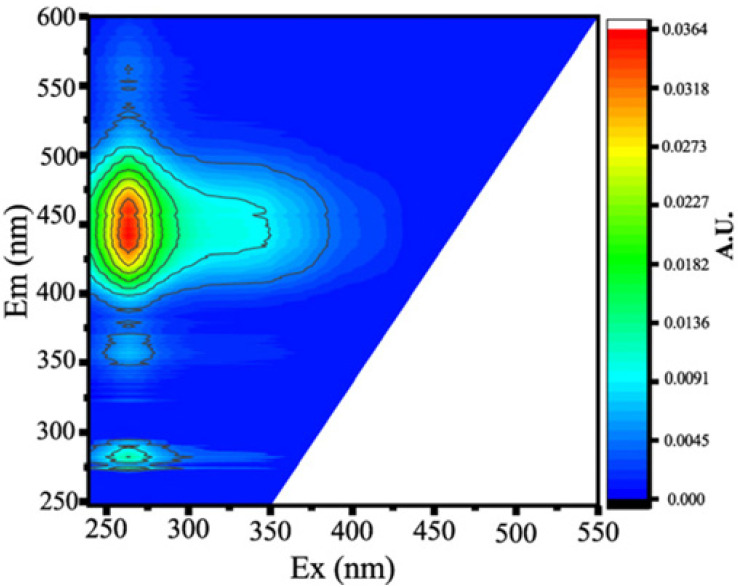	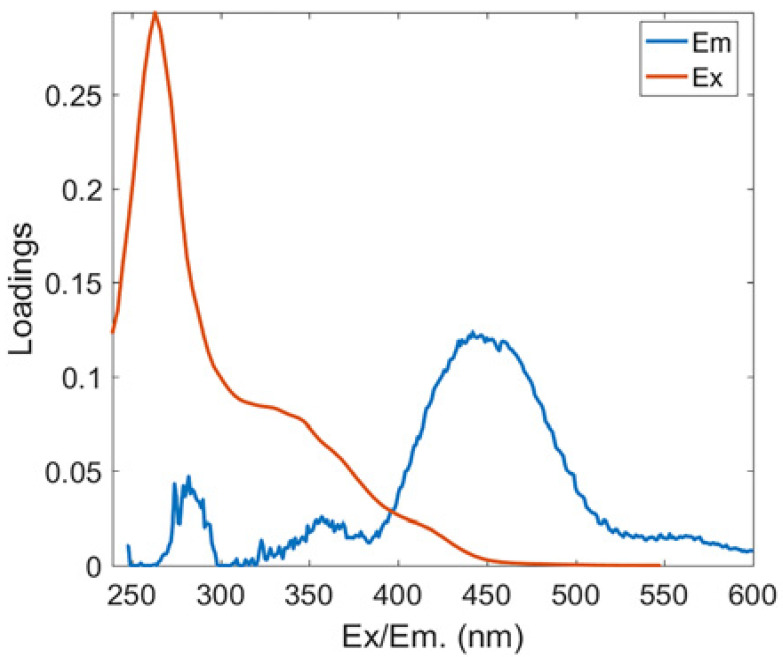	Humic-like, of terrestrial origin, poor contribution of macrophytes, allochthonous
C1: [38,39], C2: [33,40], C3 [35,36], and C4: [33]					

## Data Availability

The data that support the findings of this study are available from the corresponding author upon reasonable request.

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
