# Peer review of "Characterization of Dissolved Organic Matter from Agricultural and Livestock Effluents: Implications for Water Quality Monitoring"

_ijerph, 2023, doi:10.3390/ijerph20065121_

Round 1
Reviewer 1 Report
Generally, this is a good manuscript on providing some insight on water quality monitoring by characterizing DOM derived from agricultural effluent (AE) and livestock effluent (LE) using EEMs-PARAFAC analysis. However, there are some important queries needing confirmation before possible publication:
The abbreviation of AE and LE may confuse readers, and the full name is suggested.
Water quality of the samples can be assessed by WQI, which can be useful to quantify the water pollution.
The reasons for the sampling site should be presented, with water flow direction is suggested plot in Figure 1.
L29&L337: it is necessary to formulate the abbreviation, e.g., both C1 and C2 are protein-like component and can be used C1.
L26: “analysis” should be moved after (EEMs-PARAFAC), and “was” is suggested before “used”.
L218: Higher concentrations of what, organic or inorganic P? And than what? Such similar language description should be improved.
L227-229: Higher values of what? Why a mixed source?
L164: livestock effluent was diluted by 1000 times, which decreased DOC below 10 mg/L for fluorescence analysis. Why not dilute agricultural effluent by 10 times to below 10 mg/L for consistent analysis? In addition, more importantly, DOM in livestock effluent is 1000 times higher than that agricultural effluent, I am considering whether it is reasonable for such comparison?
L554: “components” should be used instead of “compositions”.
Figure 5: what about the p value?
Figure 6: please update the analysis procedure details of Partial Mantel test.
Many references supporting the discussion were missing, such as L378-381, L451-454, L460, etc.
Reviewer 2 Report
This paper describes an experimental approach to examine the development of using three-dimensional fluorescence to track DOM generated from AEs discharged from agricultural fields and farm discharges. The topic is interesting and suitable to the Journal as well as and the manuscript is generally well organised. However, as discussed below, the methodology developed and applied here is not properly explained to be reproduced by other researchers. As a consequence, the results obtained and the conclusions drawn are arguable. I suggest a revision of the manuscript, including additional detail and clarification regarding technical issues of the methodology and careful review and response to the comments below.
Abstract: The abstract needs more focus and needs clarity. Focus the abstract on what is new, not on general statements and provide tangible results.
Introduction: In this section the authors need to point out how this study is different from the other limited literature (in brief) since this type of work has already been studied by several previous researchers This difference will provide the motive for the study. The concept of the manuscript is not clearly developed, and at it is not entirely clear what the novelty of the manuscript is.
Materials and methods:
I think the specific section needs major improvement. The presentation of this crucial section seems like a guide for researchers (in an imperative way) rather than a scientific presentation of the overall process. Moreover, I was not able to follow the overall approach and to understand the conclusions. For example, I am not even sure how many samples were taken. Why only two sampling sites were investigated? Are these adequate enough herein?
Results and discussion.
This section is premature and seems that it has a large room for improvement. The discussion section is rather weak. Yes, we all know L.325-327 " Dynamic biogeochemical processes including microbial degradation, photochemical oxidation, and mixing can alter how much CDOM contributes to the DOM 327 pool", but arguing that " establishing holes on the surface of the sample“ is far too simple. It's much more complicated than that and therefore authors should prove this alteration based on the sources investigated in order to answer these questions.
Other comments
-Limitation and scope of the study should be provided at the end.
-The readers may wonder what are the new contributions the present research can provide?
- Please try to avoid starting sentences and paragraphs describing Tables and Figures.
- Most figures are of poor quality. Please use bigger fonts, avoid overlapping between graphs or text boxes, use appropriate captions. A detailed legend should be included with every figure and thus make it stand-alone.
- Pleas use SI units e.g. mg/L instead of mg/l
Reviewer 3 Report
In this paper, Guizhi Qi et al. studied dissolved organic matter from agricultural and livestock effluents. Generally, this paper had significant scientific and practical values and provide important results for scientific research and policymakers. Based on this point, I suggest this paper be published after several modifications. Below are my suggestions:
Major points:
(1) The English writing style is recommended to be revised by an English native speaker. In this case, the writing will be more cohesive, coherent, and clear. For example, in line 403 "In we also found an interesting phenomenon," we can understand the meaning but it is not common to be seen in the English article.
(2) The main conclusion itself needs to be enhanced. For example, in Line 34-36, "The microbial activity enhanced protein-like substances (C1 and C2) in LEs; which was associated with primary productivity." Please be clear on the details "was associated". This point is the same as the many words "influenced", and "affected". If the word can be more precise (e.g., increased, decreased), it will help the readers.
(3) The visualization looks good. However, some parts or figures are required to be improved before publication.
For example, the subtitle (a) (b) (c) (d) may be added to Figure 2 and Figure 4 to make the presentation clear. Figure 3 used (A) (B). Figure 5 used (a) (b) (c) (d) (e) (f). "Figure 5" versus "Fig. 6". Look like different authors prepare different parts. Please be consistent before publication.
Minor points:
Line 131. Figure 1. I cannot see the label "0, 2.5, 5, 10 xxxx". The font size is too small in the figure.
Line 358. Table 1. The first column “C1:272/354” is bold. "C2:272/302", "C3:308/393", and "C4:263/442" are not bold. Please check.
Line 378-Line 381. Surface waters rich in phytoplankton and algae are found in regions with strong primary production., the fluorescence of tyrosine-like B can exist either as "free" molecules or attached to proteins of algal cells or humic structures and their residues. Please check the punction of ".,"
Line 411. Figure 4. LEs-Fmax is very low in Aut. The authors should state and explain it in a clear position.
Line 519. Fig.6. Mantel's p. On my side are "< 0.01" "0.01-0.0" ">= 0.05". Do authors want to say "< 0.01" "0.01-0.05" ">= 0.05"? Please check.
Round 2
Reviewer 2 Report
The authors have performed a simple and routine work. The necessary corrections/additions, which mentioned in the first review (abstract, introduction, materials and methods etc) , did not apply in the manuscript satisfactory and I cannot find any justification for this work. In this context, the revised manuscript has been hastily finished e.g. Fig.1 and Fig.3 are given scattered (not properly aligned) in the manuscript.
